# NAPS: Natural Program Synthesis Dataset

## Abstract

We present a program synthesis-oriented dataset consisting of human written problem statements and solutions for these problems. The problem statements were collected via crowdsourcing and the program solutions were extracted from human-written solutions in programming competitions, accompanied by input/output examples. We propose using this dataset for the program synthesis tasks aimed for working with real user-generated data. As a baseline we present few models, with best model achieving 5.6% accuracy, showcasing both complexity of the dataset and large room for future research

## 1. Introduction

The task of *program synthesis* is to automatically find a program that satisfies user's specification. It is a problem that has been studied since the earliest days of artificial intelligence (Waldinger & Lee, 1969; Manna & Waldinger, 1975). With the renewed popularity of neural networks for machine learning in recent years, neural approaches to program synthesis have correspondingly attracted greater attention from the research community, which lead to great interest in datasets for program synthesis.

Most of the recent work in the field has been focused on program synthesis from examples for single domain of programming: string transformations (RobustFill (Devlin et al., 2017b), Neuro-Symbolic Program Synthesis (Parisotto et al., 2016) and Deep API Programmer (Bhupatiraju et al., 2017)) or Karel (Devlin et al. (2017a), Bunel et al. (2018)). A more domain agnostic dataset was presented in DeepCoder (Balog et al., 2016) but still featured very small programs. All of these results have crucial limitation that datasets were synthetically generated (with exceptions for small private test sets).

Recent examples of crowdsourced natural language to program datasets are WikiSQL (Zhong et al., 2017) and NL2Bash (Lin et al., 2017). Both of these datasets are also domain specific (with WikiSQL featuring only very simple version of SQL) and don't have programming concepts like variables and control flow. Django dataset (Oda et al., 2015) has very limited scope and each textual description is associated with one line of code.

Worth mentioning fully natural dataset from Magic The Gathering (Ling et al., 2016), that has natural language from people describing actions of cards and Java programs that perform this actions in the Magic environment. This dataset has very limited scope of programs, mostly requiring to figure out complex API of the environment.

Related field to program synthesis from natural language is semantic parsing: mapping of natural langauge into formal representation, which can be considered as simple programs. Recent examples of such datasets are WebQuestions (Berant et al., 2013), Overnight (Wang et al., 2015), IFTTT (Beltagy & Quirk, 2016). All of these datasets are limited to a specific sub-domain and a limited set of functional intents.

Additionally, there is work on latent program induction which does not require programs as supervision. This simplifies the dataset collection, but has limitation that programs frequently fail to generalize to different inputs (Graves et al., 2014) and does not expose interpretable program back to the user while having huge performance overhead at runtime (Kaiser & Sutskever (2015), Neelakantan et al. (2016)).

In this work we presenting Natural Program Synthesis Dataset v1.0 (NAPS), freely available at `https://goo.gl/WaBdbb`, consisting of real expert programmers' solutions for complex problems and rewritten statements in the form that is approachable at current state of technology. Dataset contains 1592 training and 455 test examples, with additional 16320 unlabelled examples for pretraining and data augmentation.

To assess the difficulty of the NAPS dataset, we implemented sequence-to-sequence and sequence-to-tree baselines. Our best model achieves accuracy of 5.6%. This shows there is a lot of room for advancement both in modeling and in data augmentation on the NAPS dataset.

---

[1]Anonymous Institution, Anonymous City, Anonymous Region, Anonymous Country. Correspondence to: Anonymous Author <anon.email@domain.com>.

Preliminary work. Under review by the International Conference on Machine Learning (ICML). Do not distribute.

*Table 1.* NAPS Dataset Structure

| Field | Description | Training A | Training B | Test |
|---|---|---|---|---|
| Solutions | Full programs in UAST format solving a competitive problem | √ | √ | √ |
| Partial solutions | Smaller pieces of the full programs | × | √ | × |
| IO examples | Input/output examples for the full programs | √ | √ | √ |
| IO schema | Input/output types and argument names for the full programs | √ | √ | √ |
| Statements | Crowdsourced problem statements in the imperative format | × | √ | √ |
| URLs | URLs to the original problem statements | √ | √ | √ |

*Table 2.* NAPS Dataset Metrics

| Metric | Training A | Training B | Test |
|---|---|---|---|
| Number of examples in the dataset | 16320 | 1592 | 455 |
| Number of examples that are partial solutions | — | 1190 | — |
| Number of synthetic statements per solution | 300 | — | — |
| Statements length, i.e. number of tokens | $173 \pm 113$ (synthetic) | $93 \pm 51$ (real) | |
| Number of lines of code per solution | $21.7 \pm 6.4$ | | |
| Number of inputs/outputs per solution | $7.5 \pm 2$ | | |

## 2. Dataset

The first release of the NAPS dataset is split into three portions. The largest dataset contains 16320 competitive problem solutions with the corresponding input/output examples and URL links to the original problem statements from the codeforces.com website from which the problem statements can be retrieved. We also accompany each solution with 300 synthetic problem statements that we used for training the baseline models, see Section 3.

The second dataset contains 1592[1] competitive programming solutions together with the partial exerts from problem solutions. Each record in this dataset is accompanied with a problem statement that was collected by the means of a crowdsourcing platform, a URL to the original problem statement, and input/output examples for non-partial solutions.

The third, smallest dataset contains 455 full problem solutions also accompanied with the crowdsourced problem statements, URLs, and input/output examples.

**Solutions:** The solutions presented in this dataset are collected from the programming competitions. We then have converted the code written in Java into our intermediate language, UAST, which additionally allowed us to unify library-specific containers and algorithms. In the future this method will also allow our models to work with solutions across programming languages such as C++, Python, C# and Pascal.

---

[1] We are currently actively expanding this dataset by running a crowdsourcing platform. This number will be updated for the camera-ready version.

**Written Statements:** We hosted a crowdsourcing platform with participants from competitive programming community, and asked them to describe the problem solution that was presented to them in UAST. The process was moderated and the participants were strongly encouraged to give descriptions that were as high-level as possible while at the same time using the language with the imperative structure of the sentences. To provide a curriculum step for the models trained on this dataset, we also asked the participants to describe smaller inner blocks of the solutions. The workers were allowed to reuse the language used for the inner blocks when describing the blocks enclosing them, but only if the larger block couldn't be described at a higher abstraction level.

**Tests:** Each full solution is accompanied with 2-10 inputs/outputs each split into two groups. The first group can be used in search or can be included into the problem specification as part of the model input. The second group can be used for the evaluation at the test time.

### 2.1. UAST Specification

UAST eliminates the burden of managing a runtime or having a compilation step. The code is convertible back and forth between UAST and Java while preserving the readability and the ability to run the input/output examples. While converting to UAST, we also remove all the file I/O and pass all the input data as arguments to the main function, and make the function return the final output. The classes are replaced with records and the class methods are replaced with global functions that accept the record as the first argument. The execution engine and tools for static and runtime analysis can be found at $URL$.

*Table 3.* UAST Specification

```
PROGRAM   ::=   {'types':  [RECORD...], 'funcs':  [FUNC...]}
                Optional record '__globals__' declares global variables. Function '__main__' is the entry point and
                optional function '__globals__.__init__' initializes the global variables.
RECORD    ::=   ['record', name, {field_name:  VAR...}]
FUNC      ::=   ['func' | 'ctor', TYPE, name, [VAR...], [VAR...], [STMT...]]
                The function entries are: the return type, the name, the arguments, the local variables, and the body.
VAR       ::=   ['var', TYPE, name]
STMT      ::=   EXPR | IF | FOREACH | WHILE | BREAK | CONTINUE | RETURN | NOOP
EXPR      ::=   ASSIGN | VAR | FIELD | CONSTANT | INVOKE | TERNARY | CAST
ASSIGN    ::=   ['assign', TYPE, LHS, EXPR]
LHS       ::=   VAR | FIELD | INVOKE
IF        ::=   ['if', TYPE, EXPR, [STMT...], [STMT...]]
FOREACH   ::=   ['foreach', TYPE, VAR, EXPR, [STMT...]]
WHILE     ::=   ['while', TYPE, EXPR, [STMT...], [STMT...]]
BREAK     ::=   ['break', TYPE]
CONTINUE  ::=   ['continue', TYPE]
RETURN    ::=   ['return', TYPE, EXPR]
NOOP      ::=   ['noop']
FIELD     ::=   ['field', TYPE, EXPR, field_name]
CONSTANT  ::=   ['val', TYPE, value]
INVOKE    ::=   ['invoke', TYPE, function_name, [EXPR...]]
TERNARY   ::=   ['?:', TYPE, EXPR, EXPR, EXPR]
CAST      ::=   ['cast', TYPE, EXPR]
TYPE      ::=   bool | char | int | real | TYPE* | TYPE% | <TYPE|TYPE> | record_name#
                The last four types correspond to an array, a set, a map, and a record type.
```

The language allows several redundancies that simplify the code analysis and the implementation of the executor and the tools. For instance, each expression has a TYPE as the second entry which eliminates the need of deducing the types. Functions require declaring local variables in advance, see Table 3. We have also introduced FOREACH and TERNARY which can be expressed through other control-flow constructs but their introduction has greatly reduced the size of the code. In addition the language is accompanied with a short library of basic functions like 'map_keys', 'string_find', etc. Full documentation on UAST can be found here *URL to be provided*.

## 3. Experimental Results

In this section we present some of our results on applying sequence-to-sequence and sequence-to-tree models for synthesizing programs from problem statements. In addition we present the data-structure that we used to perform the decoding in the sequence-to-tree model.

We train on a weighed combination of datasets A and B, with weight 10 on sampling from later to expose model to both datasets proportionally. For the dataset A we generated synthetic problem statements using a rule-based randomized method where the rules were selected to match the stylistics of the crowdsource workers as close as possible. The synthetic statements were regenerated anew at the beginning of each epoch and we include 300 synthetic statements for

each solution in the dataset A which corresponds to the number of epochs we trained our baseline models for. The evaluation was performed on the holdout dataset that did not share solutions with the training datasets.

Our sequence-to-sequence model consists of the text encoder and the program decoder mediated through the standard multiplicative attention mechanism (Luong et al., 2015). The encoder is the the bidirectional RNN with GRU cells stacked in two layers (Cho et al., 2014). The decoder is a single RNN with GRU cells augmented with a pointer mechanism (Vinyals et al., 2015). In addition to using the pointer mechanism for copying out-of-vocabulary constants and string literals from problem statements to the synthesized code, we also use it for copying in-vocabulary tokens like arithmetic operations and variable names. For this reason we preferred the soft-switch design described in See et al. (2017), which is suitable for in-vocabulary copying, over the hard-switch design described in Gülçehre et al. (2016).

### 3.1. Sequence to Tree

The sequence-to-tree model shares the same encoder and the attention mechanism with the sequence-to-sequence model but the decoding step accounts for the hierarchical nature of the program. It is done by first implementing a general purpose persistent tree data-structure (Sarnak & Tarjan, 1986) that allows storing and extending multiple UASTs

simultaneously, similarly to how it is done in Polosukhin & Skidanov (2018). The data-structure and the specific implementation of the decoder then work together where the decoder provides the nodes to extend and the data-structure extends them by forking a new tree and placing it in the priority queue based on the tree's priority, e.g. the likelihood of the entire tree defined by the logits returned from the decoder.

Each node in UAST has an access to its siblings and the parent. For each tree we store the global state of the entire tree and for each node we store two states: for its siblings and for its children. The data-structure then passes these states to the decoder which decides which of the incomplete nodes to extend based on the given states. The data-structure then handles multiple extension options for each node which is used in the search. In this paper we only provide results for the decoder that always extends the left-most incomplete node based on the global state of the tree. However this design can also easily adopt decoders from other papers, e.g. decoders described in Polosukhin & Skidanov (2018) and in Parisotto et al. (2016).

Dataset B and the Test dataset contain problem statements written by real users which poses a challenge since personal writing style varies a lot even though we tried to incentivize the consistency. The biggest challenge is the variance in the verbosity and the usage of rare words. Rules for the synthetic problem statements attempt to mimic the variance in the style but nevertheless the resulting model is still very sensitive to verbosity. Specifically, the model learns to assign a higher significance to out-of-vocabulary tokens during training than what is optimal for the test dataset.

For the sequence-to-sequence model the evaluation was performed using the beam search with the beam size equal 64. For the sequence-to-tree model the queue capacity was 64 and at each step the decoder would expand the left-most incomplete node with 64 most probable tokens yielding 64 new trees which would utilize the memory saving properties of the persistent trees. At the end we would search through the resulting 64 programs and pick the one that passed the input/output tests. The accuracy is then measured by counting the synthesized programs that pass all the input/output tests that were not used in the search. We also define 50%accuracy metric which counts the programs that pass at least 50% of the test input/output examples, see Table 4.

Interestingly, even when the model does mistakes during the inference those mistakes might be benign and it will still be passing the tests. For instance, Table 5 shows the inference example for the following problem statement:

*You are given a number var0. You have to set var2 to 2. If var0-2 is divisible by 3 you have to set var1 to 1, otherwise you have to set var1 to zero. For each var3 between 1 and*

*Table 4.* Accuracy of vanilla and pointer models with and without out-of-vocabulary copying

| MODEL | ACCURACY | 50%ACCURACY |
|---|---|---|
| VANILLA SEQ2SEQ | 0% | 0% |
| SEQ2SEQ WIHOUT OOV | 3.5% | 5.9% |
| SEQ2SEQ WITH OOV | 4.7% | 7% |
| SEQ2TREE WITH OOV | 5.6% | 7.7% |

*Table 5.* Example of the inferred program and the tests

```
int __main__(int var0)
    vars: int var1, int var2, int var3
    var2 = 2
    if (((var0 - 2) % 3) == 0)
        var1 = 1
    else
        var1 = 0
    var3 = 1
    for(; (var3 < var0); var3 = (var3 + 1))
        if (var2 < var0)
            var2 = (var2 + ((var3 * 3) + 2))
            if (((var0 - var2) ≥ 0) & ((var0 - var2) ≤ 0))
                var1 = (var1 + 1)
            else
                if (((var0 - var2) ≥ 0) & (((var0 - var2) % 3) == 0))
                    var1 = (var1 + 1)
        else
            break
    return var1
```

| Search Input | 157 | 1312861 | 6 | | | | |
|---|---|---|---|---|---|---|---|
| Search Output | 3 | 312 | 0 | | | | |

| Test Input | 26 | 152 | 158 | 4 | 71 | 3 | 155 |
|---|---|---|---|---|---|---|---|
| Test Output | 2 | 3 | 4 | 0 | 2 | 0 | 4 |

*var0-1, if var2 is less than var0 you have to, add var3\*3+2 to var2, if var0-var2 is greater than or equal to zero and var0-var2 is divisible by 3 add 1 to var1; otherwise you have to break from the enclosing loop. You have to return var1.*

Note that if $var0-var2 \geq 0$ & $var0-var2 \leq 0$ then $var0-var2 \geq 0$ & $(var0-var2)\% \ 3 == 0$. Even though the model has inferred a redundant if-clause it did not break the program's logic.

## 4. Future Work

NAPS dataset enables the program synthesis research on real-life non-trivial programs and problem statements written in a general-purpose language. The baseline metrics however demonstrate a large room for the improvement.

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
