# OpenReview forum: "NAPS: Natural Program Synthesis Dataset"
_ICML.cc/2018/Workshop/NAMPI — NAMPI 2018_

### Review · AnonReviewer1 · 2018-06-25
**A new Program Synthesis Dataset**

**Rating:** 8
**Confidence:** 5

**Review:**

Summary: This paper presents a new program synthesis dataset that presents a new large-scale challenge for the program-synthesis community. The authors construct two sets of datasets with/without partial solutions to the programming challenges obtained from codeforces.com. The code in the dataset is provided in the UAST format. The paper presents results with various seq2seq models.

Pros:
1) Presents a novel and real-world program synthesis challenge.
2) The task and the dataset is very interesting.
3) The paper is very well-written.

Cons:
1) The baselines are weak.

Questions and comments:
1) What do you think of trying a self-attention or transformer like model on this task?
2) Did you remove the comments from the code?
3) Are the solutions in the dataset officially verified solutions? Have you checked the correctness of these solutions?
4) The sentence starting from line 156-158 doesn't make sense to me: We train on a ... What is the "weight 10" there? How did you exactly combine those two datasets?

---

> ### Author Response · Authors · 2018-07-05
> **Answers**
>
> > What do you think of trying a self-attention or transformer like model on this task?
> We are actively working on complex decoders and encoders including the transformers.
>
> > Did you remove the comments from the code?
> Yes, we did.
>
> > Are the solutions in the dataset officially verified solutions? Have you checked the correctness of these solutions?
> We have validated that the solutions pass IO tests by running the UAST code using the UAST interpreter. The code of the UAST interpreter, as well as other utilities for working with the dataset is now publicly released. https://github.com/nearai/program_synthesis

---

### Review · AnonReviewer3 · 2018-06-28
**Details lacking, unclear if the dataset will be useful**

**Rating:** 5
**Confidence:** 3

**Review:**

This paper introduces a new dataset for program synthesis from text. The dataset contains multiple textual descriptions of the code (in a new language), along with a few input/output examples. Different versions of the dataset are introduced, for training, data augmentation, and evaluation. Simple seq2seq and seq2tree models obtain around 6% accuracy.

Although I think a resource like this will be quite useful, I'm not sure this is ready yet. It is difficult to evaluate the utility of the dataset since many of the details are omitted, but from what I could see from the paper and opening the files, the dataset is likely too small, too challenging, and most importantly, the text does not seem to be a natural description of the code (but seems like a literal description). Thus, I am not convinced of the utility of the dataset.

Strengths:

- As the authors point out, there is a crucial need for the community to go beyond synthesis of single statement source code from text, and a resource to enable training and evaluation of machine learning models for the task will be quite useful.

- The paper provides a good overview of the related literature, of both the models and the datasets.

- The idea of using UAST could be potentially impactful for the community, as it abstracts away many of the language-specific parts of the code, leaving a more generic structure behind. It would have been useful to know more about UAST for this reason.

- I appreciate the effort put into the included baselines; as far as I know, they are appropriate baselines for the task.

Weaknesses:

- More of my main concerns is that no description about the complexity of the code or the challenges of the text was provided. Even a few examples of the code or the problem statements would have been informative. It is unclear, till the very end, what the source code or the problem statements look like.

- Given the example at the end of the paper and looking through the dataset, it seems that the textual descriptions of the code are quite fine-grained, and describe each statement in detail. It is not clear what real-world applications are like this.

- Given the somewhat long descriptions and the size of the UAST trees, I don't think the labeled dataset is big enough. Further, the additional data, the 16k examples with synthetic rules, may be too constrained to be useful (are there major differences with and without it?).

- The three datasets seem like they're coming from very different distributions, thus making the task even more challenging. Also, would be helpful to have more clearer names than "A" and "B".

- Evaluation was only provided on the input/output pairs. Given that the code is available, an evaluation that compares the generated source code with the original one would be good.

- Writing is unclear in places, with numerous typos and grammatical mistakes. Please revise the text for the final version.

---

> ### Author Response · Authors · 2018-07-05
> **Answers**
>
> > More of my main concerns is that no description about the complexity of the code or the challenges of the text was provided. Even a few examples of the code or the problem statements would have been informative. It is unclear, till the very end, what the source code or the problem statements look like.
> We have added more examples of the statements, the original code, and the generated code in the camera-ready version.
>
> > Given the somewhat long descriptions and the size of the UAST trees, I don't think the labeled dataset is big enough. Further, the additional data, the 16k examples with synthetic rules, may be too constrained to be useful (are there major differences with and without it?).
> We are constantly expanding the dataset. Training without the dataset A results in a significant loss of generalization and, as a result, zero accuracy on the test dataset.
>
> > The three datasets seem like they're coming from very different distributions, thus making the task even more challenging. Also, would be helpful to have more clearer names than "A" and "B".
> The program code comes from the same distribution for all three files. We are constantly working on improving the process that generates synthetic descriptions for the dataset A so that it matches the other two datasets as close as possible.
>
> > Evaluation was only provided on the input/output pairs. Given that the code is available, an evaluation that compares the generated source code with the original one would be good.
> Traditional metrics, like BLEU, provide some guidance when comparing the golden and the generated program code. Unfortunately, it is not well-suited for programming tasks, a single difference in a variable name can make no significant difference to the BLEU score while at the same time it can break the entire program. Often the order of initialization of the variables with values does not matter for the program, but it can have a significant impact on the BLEU score, especially for small programs.

---

> > ### Comment · AnonReviewer3 · 2018-07-05
> > **Looking forward!**
> >
> > Thank you for the clarifying comments and plans to improve the manuscript. I look forward to the dataset and the interesting approaches it will seed!

---

### Decision · ~NAMPI_Admin1 · 2018-06-28
**Paper4 Final Decision**

Accept